# Co-Culturing Microalgae with *Roseobacter* Clade Bacteria as a Strategy for *Vibrionaceae* Control in Microalgae-Enriched *Artemia*

**DOI:** 10.3390/microorganisms11112715

**Published:** 2023-11-06

**Authors:** José Pintado, Patricia Ruiz, Gonzalo Del Olmo, Pavlos Makridis

**Affiliations:** 1Marine Ecology and Resources Group, Institute of Marine Research (IIM-CSIC), 36208 Vigo, Spain; patriciaruiz@iim.csic.es (P.R.); gdelolmo@iim.csic.es (G.D.O.); 2Department of Biology, University of Patras, 26504 Rio Achaias, Greece; makridis@upatras.gr

**Keywords:** *Artemia*, *Vibrionaceae*, vibriosis control, *Phaeodactylum tricornutum*, *Chlorella minutissima*, *Ruegeria*, *Phaeobacter*

## Abstract

Bacterial communities associated with fish larvae are highly influenced by the microbiota of live prey used as feed (rotifers or *Artemia*), generally dominated by bacterial strains with a low degree of specialization and high growth rates, (e.g., *Vibrionaceae*), which can be detrimental to larvae. Co-cultivation of microalgae used in the enrichment of *Artemia* (e.g., *Phaeodactylum tricornutum*, or *Chlorella minutissima*) with *Vibrio*-antagonistic probiotics belonging to the *Roseobacter* clade bacteria (e.g., *Phaeobacter* spp. or *Ruegeria* spp.) was studied. The introduction of the probiotics did not affect microalgae growth or significantly modify the composition of bacterial communities associated with both microalgae, as revealed by DGGE analysis. The inoculation of *P. tricornutum* with *Ruegeria* ALR6 allowed the maintenance of the probiotic in the scale-up of the microalgae cultures, both in axenic and non-axenic conditions. Using *Ruegeria*-inoculated *P. tricornutum* cultures in the enrichment of *Artemia* reduced the total *Vibrionaceae* count in *Artemia* by 2 Log units, therefore preventing the introduction of opportunistic or pathogenic bacteria to fish larvae fed with them.

## 1. Introduction

During the rearing of marine fish larvae, high mortalities are often observed which could be related to the microbiota established in the rearing system [1]. Increased knowledge of the nutritional requirements of marine fish larvae for highly unsaturated fatty acids has contributed to the development of better enrichment emulsions and thereby higher survival of the larvae [2]. Nevertheless, even in tanks originating from the same batches of fish eggs and fed the same diet high mortalities are often observed and they are often attributed to microbial imbalances [1].

The rearing of marine fish larvae is still dependent to a large degree on live feed, such as rotifers and *Artemia* [3]. Bacterial communities associated with fish larvae are highly influenced by live feed microbiota, generally dominated by bacterial strains with a low degree of specialization and high growth rates, which can be detrimental to larvae [4]. In a normal hatchery practice, before delivery to fish larvae, *Artemia* are enriched with essential fatty acids, such as polyunsaturated fatty acids (PUFAs), by feeding them with microalgae or oil emulsions [5]. Hatching and enrichment of *Artemia* involves an increase in organic matter that favours the proliferation of opportunistic bacteria [5], *Vibrio* and *Pseudomonas* spp. potentialy pathogenic to fish larvae [6]. Rinsing or disinfection of the live feed will decrease the number of bacteria, but opportunistic bacteria exhibiting a high growth rate can bloom in the fish tanks [1].

Antibacterial activity against aquaculture pathogens has been demonstrated in different microalgae species [7,8,9]. Microalgae host-specific bacterial populations and members of the *Roseobacter* clade (α-Proteobacteria) have been frequently found associated with microalgae cultures [10,11,12]. Strains belonging to this clade of the genera *Phaeobacter* (formerly *Roseobacter*, [13]) and *Ruegeria*, have demonstrated a probiotic effect due to antagonistic activity against fish pathogenic *Vibrionaceae*, preventing vibriosis in fish larvae [14,15,16]. This competitive advantage of *Roseobacter* clade bacteria has led to the study of their application as probiotics in aquaculture [10,12] as a strategy for disease control and improvement of fish larvae viability, avoiding the use of disinfectants and antibiotics. This strategy would be in line with the development of more sustainable aquaculture production by moving from the traditional “beat them” strategy commonly applied in microbial management to a “join them” approach based on ecological theory and considering microalgae bacteria interactions [4,6,7].

The administration of antagonistic bacteria has been proposed as a strategy for the control of *Vibrionaceae* bacteria during the enrichment of brine shrimp *Artemia* [6]. In a challenge trial with a model system, with axenic *Artemia* with a controlled background microbiota of four bacterial strains, the addition of the *Roseobacter*-clade bacteria, *Phaeobacter inhibens*, caused a significant reduction in the growth of introduced *Vibrio anguillarum*, that reached levels 3–4 log lower than in the control [17]. However, *Roseobacter*-clade bacteria are not predominant in *Artemia* and the approach implies the regular administration of the probiotic. Alternatively, pre-culturing *Vibrio*-antagonistic bacteria with the microalgae used in the enrichment of *Artemia* would have the advantage of the probiotic to be co-cultivated with the microalgae, potentially maintaining in the culture and simplifying the application. Moreover, co-cultivation can benefit of mutualistic interactions between bacteria and microalgae. It has been demonstrated that the antagonistic activity of *Roseobacter*-clade bacteria (e.g., *Sulfitobacter* sp.) against *Vibrio anguillarum* is enhanced by the metabolites excreted by microalgae e.g., *Chlorella vulgaris* or *Nannochloropsis oculata* [7,8].

The feasibility of co-culturing microalgae with beneficial bacteria has been demonstrated by several authors and the effectiveness of such an approach, as a way to deliver probiotics to live prey (e.g., *Artemia*) and control pathogens, has been assessed [6]. However, microalgae bacteria interactions are complex and different bacterial strains may have specific interactions with different microalgae species [18].

The aim of this work was to study the co-culture of different antagonistic bacterial strains, belonging to the genus *Phaeobacter* and *Ruegeria*, with two microalgae commonly used for the enrichment of prey (e.g., *Artemia*) in fish larviculture: *Phaeodactylum tricornutum* (Bacillariophyta) and *Chlorella minutissima* (Chlorophyta). The combination of the different bacteria with the two microalgae permits the determination of potential specificity and selection of the most effective combination, as well as to study the possible modification of the bacterial communities associated with the microalgae.

In marine fish hatcheries, microalgae are produced in batch or continuous cultures in non-axenic conditions. Batch cultures, with several scale-up steps from small volumes of initial axenic cultures in test tubes or Erlenmeyer flasks, to thousands of liters of non-axenic cultures in indoor or outdoor tanks or plastic bags, are the most reliable and easiest culture method [19]. Batch cultures are, therefore, the most widely used system for mass production in aquaculture production units.

The hypothesis of the present work was that early colonization of the microalgae with selected vibrio-antagonistic bacteria would assure the maintenance and delivery of these probiotic bacteria in the scale-up in non-axenic conditions. Mixed cultures of the antagonistic bacteria and microalgae could be used as a strategy for bacterial control, preventing colonization of *Artemia* sp. by opportunistic or potentially pathogenic bacteria, reducing *Vibrionaceae* load during the enrichment process and, thus, improving survival in fish larvae fed with those prey organisms.

## 2. Materials and Methods

### 2.1. Microalgae Strains and Cultures

Non-axenic microalgae *Phaeodactylum tricornutum* Bohlin and *Chlorella minutissima* strains were obtained from the IIM-CSIC and the HCMR collections, respectively. Axenic *P. tricornutum* strain CCMP630 was obtained from the Bigelow National Center for Marine Algae and Microbiota (East Boothbay, ME, USA).

Microalgae were cultivated in Guillard’s [20] F/2 medium (Cell-Hi F2P Varicon Aqua, Malvem, UK) at 0.30 g·L^−1^ in seawater. In the case of *P. tricornutum*, the medium was supplemented with 30 mg·L^−1^ of silicate (F/2-S). Microalgae pre-culture was conducted in 300 mL flasks with 100 mL sterile medium inoculated at 5% (v/v) with a 7-day-old culture. For the scale-up, after 7 days, 100 mL of the pre-cultures were used to inoculate 6 L flasks with 5 L of sterile culture medium. Then, 7 days later, the whole content of the flask was used to inoculate plastic bags with 50 L of non-sterile medium and cultured for 7 days. Scale-up of *P. tricornutum* cultures was conducted with axenic and non-axenic microalgae strains. Algae cells were counted by microscopy, using a haemocytometer.

### 2.2. Bacterial Strains and Cultures

The bacterial strains *Phaeobacter gallaeciensis* CECT7277, and *Phaeobacter inhibens* CECT7251 were obtained from the Spanish Type Culture Collection (CECT, Valencia, Spain). *Ruegeria* sp. LRC4 and *Ruegeria* sp. ALR6 were isolated at IIM-CSIC from turbot larvae cultures based on their antagonism against *V. anguillarum* [21]. All the strains were kept at −80 °C in Marine Broth (MB, Difco 2219) with glycerol (at a final concentration of 15%) and routinely cultured on MB. Bacterial strains were re-activated in 4.5 mL of MB by incubation at 20 °C for 72 h in the dark. One mL of the pre-culture was used to inoculate 100 mL of MB as previously described [14] and cultured in Marine Broth (MB) at 20 °C for two days. An adequate volume of culture was added to sterile-filtered supernatants of algae cultures or was used to inoculate algae cultures at an initial concentration of 10^5^ colony-forming units (CFU) mL^−1^.

### 2.3. Culture of Bacterial Strains in Microalgae Supernatants

For the detection of potential negative or positive effect of the microalgae cultures on the *Roseobacter*-clade bacteria used in co-cultures, the *P. gallaeciensis* and *Ruegeria* strains were cultured in sterile-filtered (0.22 µm) spent medium of 7 days cultures of *P. tricornutum* or *C. minutissima*. Control cultures were conducted in parallel on MB and F/2 medium.

### 2.4. Co-Culture Trials in Plates

Combinations of the 2 microalgae with the 4 bacteria were co-cultured in sterile 24-well dishes with 2 mL of SSW and 0.5 mL of a two-week-old culture of the microalgae (10^6^–10^7^ cells mL^−1^). Then, 10 µL of a 2-day-old bacterial culture was used to inoculate the algae cultures at an initial concentration of 10^5^ CFU mL^−1^. Nutrients (F/2 or F/2-S medium) were added at a final concentration of 0.30 g L^−1^ at 0 and 7 days after inoculation, and evaporation was compensated by the regular addition of sterile deionised water. Control cultures of the microalgae without inoculated bacteria were conducted in parallel. Two replicates were done for each condition. The plates were kept under daylight illumination and gentle agitation in an orbital platform shaker (Unimax, Heidolph, Germany) at room temperature (20–25 °C).

### 2.5. Scale-Up of the Selected Co-Culture

*P. tricornutum* and *Ruegeria* sp. ALR6 was the selected combination, based on the results of the co-culture trials (see Results Section 3.1). *P. tricornutum* cultures were conducted, in duplicate, with axenic and non-axenic strains. Pre-cultures were conducted in 300 mL flasks with 100 mL of F/2-S sterile medium inoculated at 5% (v/v) with a 7-day-old culture. *Ruegeria* sp. ALR6 was cultured in Marine Broth at 20 °C for two days and was used to inoculate microalgae pre-cultures at an initial concentration of 10^5^ CFU mL^−1^. Controls were conducted in parallel with microalgae without the addition of bacteria.

After 7 days, 100 mL of each of the microalgae pre-cultures were used to inoculate 6 L flasks with 5 L of sterile culture medium. Then, 7 days later, the whole content of the flask was used to inoculate plastic bags with 50 L of non-sterile medium and cultured for 14 days. Cultures were conducted under constant daylight illumination and aeration in a temperature-controlled room at 20 °C.

### 2.6. Artemia Enrichment

The four resulting *P. tricornutum* cultures were used in the 24 h enrichment of *Artemia metanauplii*. Decapsulation of *Artemia salina* cysts (INVE) was performed using 15 mL of a solution of 0.5 g of active hypochlorite and 0.15 of sodium hydroxide in tap water, per gram of cysts. After decapsulation, the cysts were concentrated by filtration through a 150 µm mesh, neutralized with hydrochloric acid (0.1 N) and thoroughly washed with flowing seawater. The *Artemia* cysts were incubated for 24 h at 29 °C, in seawater (35‰ of salinity), under light and continuous aeration.

Freshly hatched nauplii were concentrated on a 125 µm mesh, rinsed and maintained for 30 min in running tap water. Afterwards, the nauplii were transferred (about 100 nauplii mL^−1^) to 5 L buckets containing 4 L of seawater and 1 L of the corresponding culture of *P. tricornutum* with 10^6^ cells mL^−1^ and maintained for 24 h at 27 °C with gentle aeration.

### 2.7. Microbial Analysis

Samples from microalgae cultures were taken at 0, 2, 7, and 15 days. Samples were also taken from *Artemia* enrichment water and enriched *Artemia nauplii* at 0, 1 and 4 days. *Artemia nauplii* were visually counted in triplicate by taking 1 mL of *nauplii* suspension in a glass pipette, and the adequate volume for 1000 samples of *Artemia nauplii* was filtered through a 125 µm mesh, washed with sterile seawater and homogenized for microbiological analysis.

Total bacteria concentration in microalgae culture was analysed by serial dilutions in seawater and plating on MA. Introduced *Roseobacter* bacteria were identified in the MA plates by the characteristic brown pigmentation of the colonies [21]. Dilutions were also plated on TCBS-Agar medium for the estimation of total *Vibrionaceae* concentration. Plates were incubated at 20 °C for 7−10 days and CFUs were counted.

### 2.8. Denaturing Gradient Gel Electrophoresis (DGGE)

DGGE was performed to analyse the possible changes in bacterial communities associated with the microalgae, induced by the introduced bacteria. Aliquots (1.5 mL) of the microalgae cultures were centrifuged for 10 min at 3200× *g*. The microalgae were re-suspended on sterile seawater, centrifuged again in the same conditions and the pellet was kept at −20 °C for further DNA extraction. PCR-DGGE [21] was conducted using the Bio-Rad DCode apparatus following a procedure based on Muyzer et al. [22]. Fragments amplified were loaded on 8% (wt/vol) polyacrylamide gels in 1X TAE with 30 to 60% gradient urea-formamide (100% corresponded to 7 M urea and 40% [v/v] formamide) increasing in the direction of electrophoresis. A total of 500 ng of the PCR product were loaded on the gel. A control with 250 ng of PCR product from DNA extracted from pure cultures of the *Roseobacter*-clade strains was included. All parallel electrophoresis was performed at 60 °C. Gels were run for 10 min at 20 V and 3 h at 200 V, stained with ethidium bromide for 10 to 15 min and rinsed for 20 to 30 min in distilled water. The DGGE profiles obtained were subsequently processed using Quantity One v4.4.1 software package (Bio-Rad, Hercules, CA, USA).

### 2.9. Analysis of DGGE Gels

DGGE gel images were converted, normalized, and analysed by using the BioNumerics 7.1 software (Applied Maths) following the manufacturer’s instructions. The community fingerprints’ similarities between samples were determined using the Dice similarity coefficient (D_sc_). A band position tolerance of 1% and optimization of 0.5% were applied for the analysis using the Dice coefficient. For the analysis, the bands corresponding to the chloroplasts of the algae and the inoculated bacteria were excluded to avoid their influence. This allowed us to create dendrograms based on the presence or absence of bands, utilizing the unweighted data and UPGMA as the clustering algorithm. The similarity matrix generated in BioNumerics software was employed to create a Multidimensional Scaling (MDS) plot in RStudio (version 4.2.2) for the experiment of co-culture of microalgae with different bacteria. The classical multidimensional scaling (CMDS) was carried out using the cmdscale() function from the ‘stats’ package (version 3.6.2), and the resulting visualization was crafted using ‘ggplot’ from the ggplot2 package. This approach was used to discern the sample groupings based on their similarity percentages.

### 2.10. Statistical Analysis

A confidence interval (*C.I.*) considering the standard deviation was calculated with 95% confidence for each sample (microalgae and bacteria counts) at each point analyzed and pairwise comparisons were made. The *C.I.* was calculated as indicated below:C.I.=X¯±Z·σn
where $\bar{X}$ is the mean, Z is the Z-value corresponding to the confidence level chosen (in this case 1.96), *σ* the standard deviation and n the sample size. More information can be found in the next link: https://www.calculator.net/confidence-interval-calculator.html, accessed on 26 October 2023.

It was checked whether in these comparisons the confidence intervals of each sample overlapped, which would indicate that the observed differences would not be significant and vice versa. Only the last days of the different experiments were compared in order to simplify the analysis.

## 3. Results

### 3.1. Co-Culture of Bacterial Strains and Microalgae in Plates

No stimulation or inhibition of growth was observed for any of the four *Roseobacter* clade bacteria cultured in both algae supernatants (Figure 1). Similarly, as for the control in SSW, bacteria concentration in algae supernatants increased 1 Log in the first two days and remained stable afterwards, for at least 7 days.

In co-cultures, the introduced *Roseobacter*-clade bacteria had no effect on algae growth, or total bacteria concentration (Figure 2). Introduced bacteria maintained initial concentration for at least 15 days in all cases except for *Ruegeria* sp. ALR6 in *P. tricornutum* cultures, in which 2 Log growth of *Ruegeria* was observed.

Regarding the bacteria inoculated at time 15, only the *Ruegeria* strains showed significantly (their C.I. did not overlap, Appendix A) to be in a higher concentration in the *Phaeodactylum* cultures than in the *Chlorella* cultures.

DGGE analysis revealed that the introduction of the *Roseobacter*-clade bacteria did not modify the composition of bacterial communities associated with the microalgae cultures. Bands corresponding to added *Phaeobacter* strains were not detected in the gels (Appendix A) in both microalgae, most likely due to detection limitation below 10^6^ CFU [21].

The dendrograms generated using Dsc and UPGMA clustering, with samples at the beginning and the end of the culture (0 and 15 days, respectively), indicated that among the samples belonging to the microalgae *C. minutissima* (Figure 3), it was observed that they were distributed in the dendrogram more according to time, while the inoculation of the bacteria did not seem to affect the profile of the DGGE bacterial communities. This was also support by MDS analysis (Appendix A), were samples at day 15 grouped closer than those at day 0. On the contrary, the samples belonging to *P. tricornutum* (Figure 4) did seem to show a distribution according to the presence of the inoculated bacteria, those samples on one hand being more similar with inoculated bacteria belonging to the genus *Ruegeria*, and on the other hand, those with inoculated bacteria of the genus *Phaeobacter*. The MDS analysis (Appendix A) showed similar results, however grouping on the basis of the inoculum added is not clear, suggesting that, for *Phaeoadactylum*, bacterial communities are more similar between samples than in the case of *Chlorella*.

### 3.2. Scale-Up of P. tricornutum and Ruegeria sp. ALR6 Co-Culture

Introduced probiotics did not affect negatively the algae growth in both axenic and non-axenic strains (Figure 5). In the first step, bacteria introduced in the flasks grew in parallel with the algae, reaching in two days a maximum concentration, of over 10^7^ CFU mL^−1^. Both cultures, axenic and non-axenic, performed similarly in the scaling up to 5 L and, after, to 50 L, reaching maximal algae biomass after 7 days in the 50-L bags (21 days from the beginning of the culture). At day 21, the number of *Ruegeria* sp. ALR6 per *P. tricornutum* cell was in the axenic culture 1 Log higher than in the non-axenic (6.3 +/− 0.079 and 5.2 +/− 0.147 Log CFU mL^−1^, respectively), being this different significantly according to the C.I. calculated at this point of the scale up for Control and inoculated culture (Appendix A).

The DGGE profiles detected the presence of a band corresponding to the introduced *Ruegeria* sp. ALR6 strain during the whole scaling-up process, both with axenic and non-axenic *P. tricornutum* (Appendix A). Regarding the dendrograms generated using Dsc and UPGMA clustering (Figure 6) with samples at the beginning and the end of the culture (0 and 21 days, respectively), it seems that it is the type of microalgae (axenic and non-axenic) that determines the similarities in the microbial community profiles. This is especially noticeable at time 0, where the samples in general have a percentage of similarity with the rest of the times below 40%, except for non-axenic microalgae control, where the similarity between days 0 and 21 was 54.5%.

### 3.3. Artemia Enrichment

In the enrichment of *Artemia* with the microalgae, the concentration of *Ruegeria* sp. ALR6 in the water for the added initially axenic or initially non-axenic microalgae cultures was 10^5^ and 10^4^ CFU, respectively. Those values were maintained for at least the four days of the enrichment. The presence of *Ruegeria* sp. ALR6 reduced by 2 Log units total *Vibrionaceae* in *Artemia* on the first day (Figure 7) and those reduced levels, with respect to controls, were maintained for at least four days in the case of *Artemia* enriched with initially axenic *P. tricornutum* co-cultured with *Ruegeria* (this reduction was significant for both cases according to the C.I. calculated for the different conditions at day 4, Appendix A).

According to the dendrogram results of the DGGE profiles (Figure 8), the bacterial community of *Artemia* enriched for 48 h with *P. tricornutum* showed a 70.1% similarity profile between *P. tricornutum* with or without *Ruegeria* sp. ALR6. The percentage of similarity between *Artemia* fed with non-inoculated *P. tricornutum* (Control) CCMP and IIM was 80.0%, while those fed with the microalgae inoculated with the bacterium was 88.9% between the CCMP and IIM strains. This could indicate that the bacterial profile obtained from an axenic microalgae (CCMP) from a non-axenic microalgae (IIM) varies between 20 and 10% similarity and also, that the introduction of *Ruegeria* sp. ALR6 in both cases, does not significantly modify the bacterial communities associated with *Artemia*, although it has an effect in total *Vibrionaceae*.

## 4. Discussion

In marine aquaculture, microalgae are used in the rearing of marine fish larvae either for the enrichment of live feed organisms or are added directly to the fish tanks during the “green water” technique. In both cases, microalgae may have a positive effect on the microbial balance of the rearing system. Members of the *Vibrionaceae* family are often observed in high numbers in fish larvae and may cause disease to the larvae [23,24]. The use of live microalgae in the rearing tanks of fish larvae or in the enrichment of live feed (e.g., *Artemia* spp.) may inhibit the growth of opportunistic bacteria through either the production of antibacterial compounds by the microalgae cells [7] or by the antibacterial activity of microbial populations associated with the microalgae cultures [6]. Bacteria isolated from microalgae cultures have shown inhibitory activity against fish pathogens [25].

The present work explored the colonization of microalgae with selected vibrio-antagonistic bacteria from the *Roseobacter* clade as a strategy for bacterial control, preventing colonization of *Artemia* sp. during the enrichment process by opportunistic or potentially pathogenic bacteria.

In a simulation of the enrichment process of *Artemia* in microalgae added *Ruegeria* sp. ALR6 (5–6 Log CFU mL^−1^), it was shown that presumptive *Vibrionaceae* were kept 10 to 100 times lower than in the control treatment where no *Ruegeria* sp. ALR6 was added. This meant that this approach could be quite effective in keeping *Vibrionaceae* at a low level, which could be crucial for the survival of the fish larvae as *Vibrionaceae* often include genera that can be potentially harmful to fish, including pathogenic or opportunistic bacteria.

However, when co-culturing microalgae with specific bacteria, several aspects shuld be previously considered. Microalgae cells may influence the bacterial population in their microenvironment by the production and excretion of various compounds, which inhibit or promote these bacterial populations [18,26,27,28]. The two microalgae species used in this study (*Phaeodactylum tricornutum* and *Chlorella minutissima*) did not seem to produce any inhibitory compounds for the growth of the bacteria tested (Figure 1). Members of the *Roseobacter* clade are often found associated with microalgae cultures, so we could assume that these bacteria can co-exist with microalgae cells and are not influenced negatively by compounds produced by the microalgae or other environmental conditions present in microalgae cultures such as high oxygen concentrations.

In turn, bacterial populations may produce compounds such as vitamins that favour the growth of microalgae, or on the contrary, may act as pathogens for the microalgae cells [29]. In general, bacteria decompose microalgae exudates, release nutrients in the water and thereby increase the growth of microalgae. In dense algal cultures, the total numbers of bacteria generally increase and this was also observed in our experiments, where the total bacterial load increased by or remained stable in algal cultures (Figure 2 and Figure 5).

*Roseobacter* bacteria promote algal growth by biosynthesizing growth stimulants (e.g., auxins) or by secreting antibiotics that can give protection against pathogens [30,31]. However, in response to some circumstances, such as the ageing of the microalgae, bacteria of the genus *Phaeobacter* can produce algaecides, the roseobacticides, which cause lysis of microalgae cells. This dual behaviour, switching from mutualistic to opportunistic pathogen has been reported in *Emiliania huxleyi* [30]. It has been shown in a previous study that *P. inhibens* and *P. gallaeciensis* produce roseobacticides, whereas *Ruegeria* strains did not produce them [32]. In this study, no negative effect of the addition of these bacteria was shown in cultures of the two microalgae species tested.

D’Alvise et al. 2012, demonstrated that *Phaeobacter gallaeciensis*, could effectively establish in axenic cultures of the microalgae *Tetraselmis suecica* and *Nannochloropsis oculata*, and in rotifers, preventing Vibriosis in cod larvae, due to the production of tropodithietic acid (TDA) [14]. Moreover, in an experiment with non-axenic *Artemia* and *Teraselmis suecica* cultures obtained from an aquaculture unit, inoculated with *Vibrio anguillarum* (10^4^ CFU mL^−1^), the addition of *Phaeobacter inhibens* (10^4^ CFU mL^−1^) promoted a reduction of *V. anguillarum* by 3 Log units [17]. However, that research did not analyse the influence of the probiotic on bacterial communities of the hosts. In our experiments, we found that the addition of *Ruegeria* ALR6 did not significantly modify the composition of bacterial communities associated with the *C. minutissima* and *P. tricornutum* cultures (Figure 3 and Figure 4). This could be explained by the fact that the compounds produced by *Ruegeria* sp. ALR6 may antagonize specific bacterial groups present in fish cultures (e.g., *Vibrionaceae*) but do not affect bacteria commonly associated with microalgae that may have evolved to be resistant to those compounds. Phytoplankton species (e.g., *Nannochloropsis oculata*) enhance the ability of *Roseobacter*-clade bacteria to inhibit *Vibrio anguillarum* [7]. Furthermore, the *Roseobacter*-clade bacteria incorporate more efficiently than *V. anguillarum* the photosynthetic metabolites produced by the microalgae. This cooperation between microalgae and bacteria may also occur with the microalgae species used in the present study and should be further studied.

One of the main concerns associated with bacterial manipulation in the culture of organism, such as microalgae, is the potential emergence of adverse interactions or disbalances of bacterial communities, moreover, when the probiotic effect is based on antagonisms agains potential pathogens, as is this case. As previously mentioned, our experimentation involving both bacteria and microalgae did not yield any observable deleterious effects. The possible alteration of the microbial communities associated with the culture system due to the introduction of the bacteria should be considered, in order to ensure that probiotics only have an antagonistic effect against the detrimental microorganisms and not the potentially beneficial ones, which could lead to dysbiosis phenomena [33].

In our study, the analysis of DGGE profiles (Figure 3, Figure 4, Figure 6 and Figure 8), based on similarity percentages, suggested that the inoculation of the bacteria did not produce significant dissimilarity among the sampled microbial communities. Consequently, we can infer that the administration of *Ruegeria* sp. ALR6 did not exert a discernible impact on the overall microbiota composition within microalgae cultures. Rather, its antagonistic effect appears to be constrained primarily to specific taxonomic groups, such as *Vibrionaceae* (Figure 7) as discussed below.

This study contributes to the application of microbial management in marine fish larviculture by adding probiotic bacteria to the microalgae cultures during the up-scaling process. Our results showed that the added bacteria, as they belong to strains that are part of the normal microbiota of microalgae cultures, showed stability and remained viable for a long period in large-scale cultures (Figure 5). These bacteria-colonized microalgae can be introduced in the rearing process either as “green water”, or as part of the enrichment of live feed before addition to the fish tanks, without changing the profiles of the microbial communities present in the microalgae cultures (Figure 6).

In our work, we monitored total *Vibrionacae* in rinsed *Artemia*, using a procedure that simulates the one used in hatcheries, and therefore we can demonstrate the enrichment of *Artemia* with the microalgae and the probiotic reduced from 10^4^ to 10^2^ the CFU of total *Vibrionaceae* per *Artemia*, maintaining similar levels of total bacteria per *Artemia* around 10^5^ CFU. These results mean that the antagonistic bacteria *Ruegeria* sp. ALR6 inhibits the initial proliferation of *Vibrionaceae*, which occurs in the enrichment process most likely due to the availability of nutrients from which fast-growing bacteria (*r*-strategist [34]) take advantage. As was observed in the controls, even though total bacteria continued to increase by one Log unit from day 1 to day 4, total *Vibrionaceae* maintained almost constant levels or reduced slightly. This may be due to nutrient depletion, which may favour the maintenance of biofilm-forming bacteria, such as those from the *Roseobacter*-clade and would give an advantage to the introduced *Ruegeria* sp. ALR6.

In addition to the pathogen antagonistic effect, *Roseobacter*-clade bacteria can improve microalgae growth by providing vitamins or growth factors. Mutualistic behaviour has been studied in a model microbial system consisting of diatom *Thalassiosira pseudonana* and bacterium *Ruegeria pomeroyi* DSS-3 [35]. The bacterium provides vitamin B12 required by the diatom, while the heterotrophic bacterium is dependent on organic carbon and a usable nitrogen source produced by the diatom. Moreover, the presence of the *Ruegeria* bacterium triggered differential expression of genes in the diatom, modulating the production of several metabolites supporting the bacterial growth [32]. These mutualistic interactions could also have implications in the use in aquaculture of microalgae enriched with *Roseobacter*-clade bacteria and should be further investigated.

## 5. Conclusions

In conclusion, inoculation of *P. tricornutum* with *Ruegeria* ALR6 allows the maintenance of the probiotic in the scale-up of the microalgae cultures, both in axenic and non-axenic conditions. Using *Ruegeria*-inoculated *P. tricornutum* cultures in the enrichment of *Artemia* reduces the total *Vibrionaceae* in that prey, which will prevent the introduction of opportunistic or pathogenic bacteria to fish larvae fed with them. To substantiate these findings, it would be interesting to conduct challenge trial with fish larvae infected with *Vibrio* spp. [14]. These assays would provide an in-vivo verification of the observed probiotic effect of microalgae enriched with *Roseobacter*-clade bacteria and further demonstrate its potential in controlling fish-pathogenic organisms.

Microalgae enriched with the probiotic *Roseobacter*-clade bacteria could also be delivered directly to the larvae rearing tanks (the “green water technique”) [6], boosting the bacterial control effect and enhancing the survival and growth of fish larvae. This approach would ensure the maintenance of the probiotics in the rearing systems.

The use of *Roseobacter* bacteria could also be useful in the production of microalgae concentrates as microalgae paste, which is frequently used in marine fish hatcheries to save energy and personnel costs. Having those bacteria in the microalgae paste could provide a *Vibrionaceae*-control effect in *Artemia* enrichment, and this methodology should be explored further.

## Figures and Tables

**Figure 1 microorganisms-11-02715-f001:**
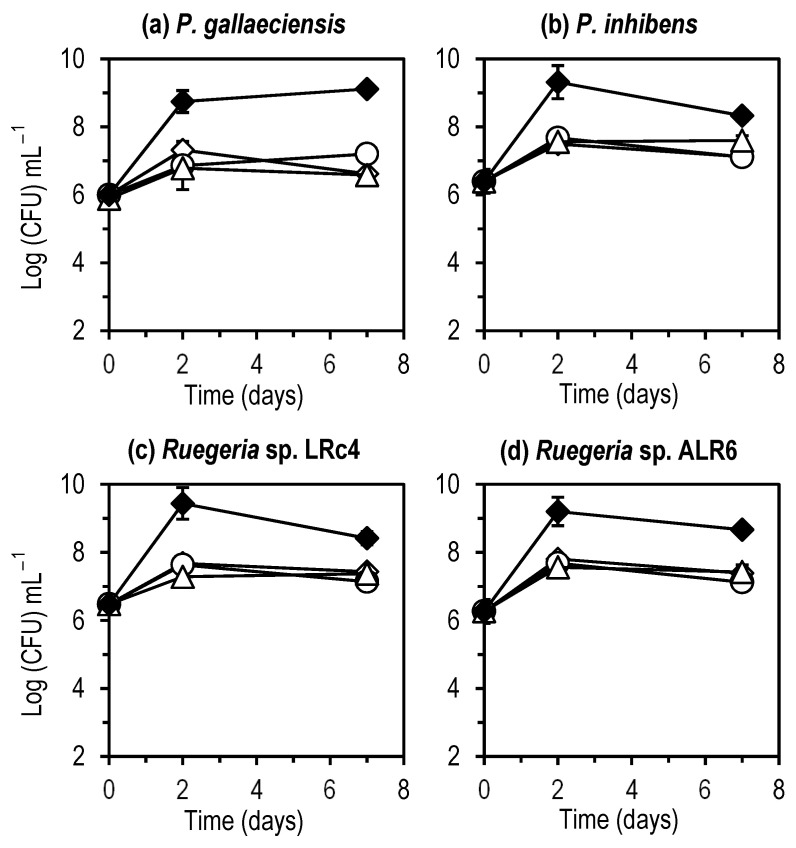
Cultures of the bacteria *Phaeobacter gallaeciensis* (**a**), *Phaeobacter inhibens* (**b**), *Ruegeria* sp. LRc4 (**c**), and *Ruegeria* sp. ALR6 (**d**) in Marine Broth (◆), supernatants of the algae *Phaeodactylum tricornutum* (◇) and *Chlorella minutissima* (○), and in Seawater with F/2 medium (△). Each point represents the average (*n* = 2) of the concentration (in Log (CFU) mL^−1^) of the four bacteria in the different mediums. Error bars represent the standard deviations. The graph was made with Microsoft Excel 2019.

**Figure 2 microorganisms-11-02715-f002:**
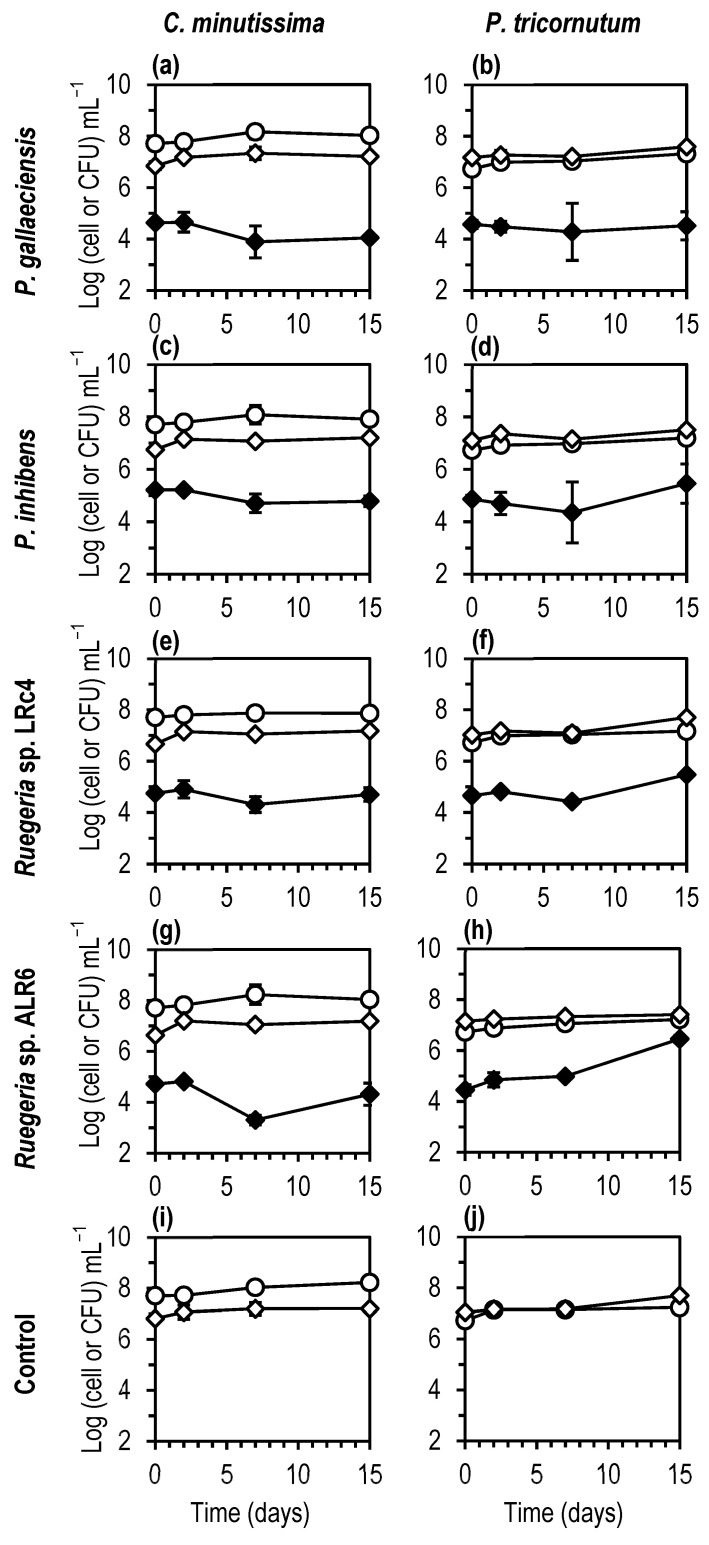
Mixed cultures of the algae *Phaeodactylum tricornutum* (**a**,**c**,**e**,**g**,**i**) and *Chlorella minutissima* (**b**,**d**,**f**,**h**,**j**) with the bacteria *Phaeobacter gallaeciensis* (**a**,**b**), *Phaeobacter inhibens* (**c**,**d**), *Ruegeria* LRc4 (**e**,**f**) and *Ruegeria* ALR6 (**g**,**h**) and Control cultures of algae with no addition of bacteria (**i**,**j**). (○) Algae cells (Log (cells) mL^−1^); (◇) Total bacteria (Log (CFU) mL^−1^); (◆) Introduced *Roseobacter* bacteria (Log (CFU) mL^−1^). Each point represents the average (*n =* 2) of the concentration (in Log (cells or CFU) mL^−1^) of the microalgae and bacteria. Error bars rep the standard deviation. The graph was made with Microsoft Excel 2019.

**Figure 3 microorganisms-11-02715-f003:**
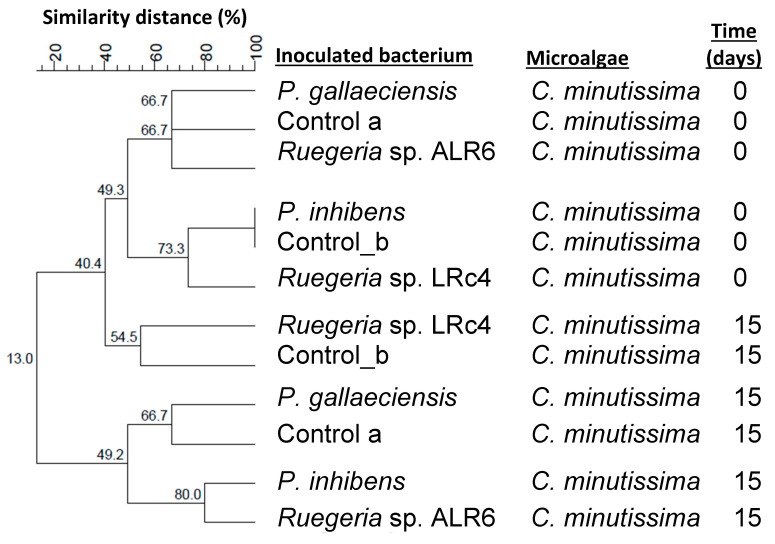
Dendrogram of DGGE profiles of bacterial communities in the co-culture of bacterial strains and *Chlorella minutissima* in plates samples. The dendrogram was elaborated based on Dice’s similarity coefficients and the clustering performed by the UPGMA algorithm included in the BioNumerics v7.1 software. The samples analyzed are the cultures of the microalgae *C. minitissima* inoculated with the four bacteria with antagonistic activity (*Phaeobacter gallaeciensis*, *Phaeobacter inhibens*, *Ruegeria* sp. LRc4 and *Ruegeria* sp. ALR6.) plus the control samples without inoculation, at different times (0 and 15 days). The graph was made by BioNumerics 7.1 software.

**Figure 4 microorganisms-11-02715-f004:**
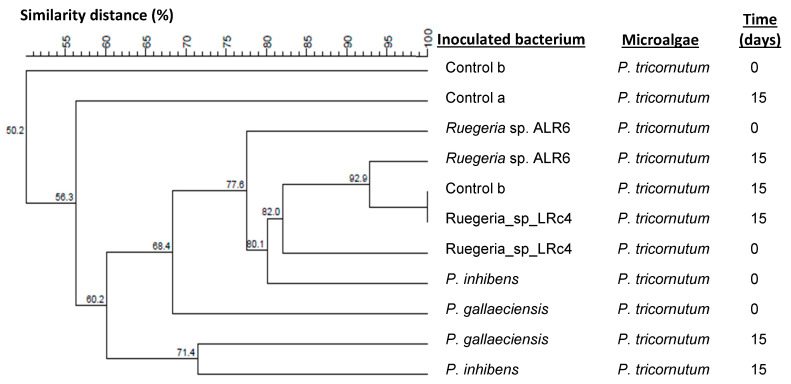
Dendrogram of DGGE profiles of bacterial communities in the co-culture of bacterial strains and *Phaeodactylum tricornutum* in plates samples. The dendrogram was elaborated based on Dice’s similarity coefficients and the clustering performed by the UPGMA algorithm included in the BioNumerics v7.1 software. The samples analyzed are the cultures of the microalgae *P. tricornutum* inoculated with the four bacteria with antagonistic activity (*Phaeobacter gallaeciensis*, *Phaeobacter inhibens*, *Ruegeria* sp. LRc4 and *Ruegeria* sp. ALR6.) plus the control samples without inoculation, at different times (0 and 15 days). The graph was made by BioNumerics 7.1 software.

**Figure 5 microorganisms-11-02715-f005:**
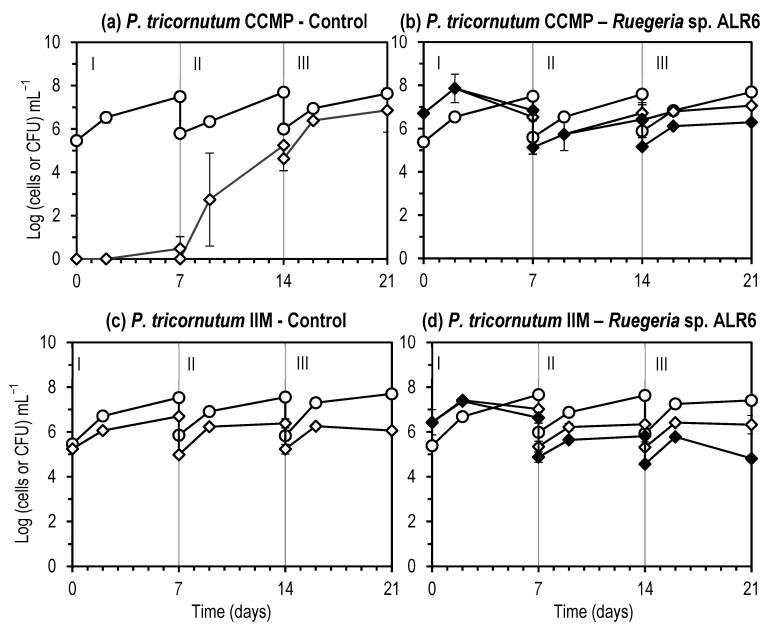
*Phaeodactylum tricornutum* cultures of axenic (CCMP, (**a**,**b**)) or non-axenic (IIM, (**c**,**d**)) strains inoculated with *Ruegeria* sp. ALR6 (**b**,**d**) or not inoculated (Control, (**a**,**c**)), scaled-up from (I) 100 mL flask to (II) 5 L flask and to (III) 50 L bags. (○) *P. tricornutum* Log cells mL^−1^; (◇) Total bacteria Log CFU mL^−1^; (◆) Introduced *Roseobacter* bacteria (*Ruegeria* sp. ALR6, in Log CFU mL^−1^). Each point represents the average (*n =* 2) of the concentration (in Log (cells or CFU) mL^−1^) of the microalgae and bacteria. Error bars represents the standard deviation. The graph was made with Microsoft Excel 2019.

**Figure 6 microorganisms-11-02715-f006:**
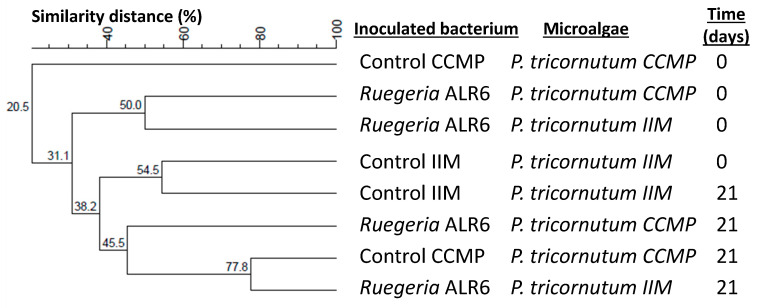
Dendrogram of DGGE profiles of bacterial communities in the scale-up experiment samples. The dendrogram was elaborated based on Dice’s similarity coefficients and the clustering performed by the UPGMA algorithm included in the BioNumerics v7.1 software. The samples analyzed are the scale-up of the cultures of the two *P. tricornutum* (CCMP and IIM) inoculated with *Ruegeria* sp. ALR6 and the control samples without inoculation, at different times (0 and 21 days). The graph was made by BioNumerics 7.1 software.

**Figure 7 microorganisms-11-02715-f007:**
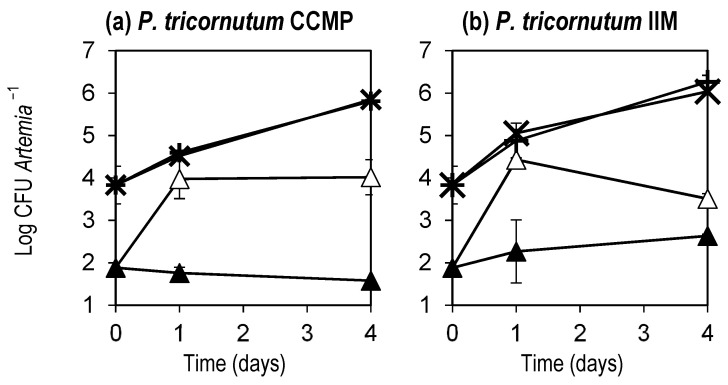
Total bacteria and total *Vibrionaceae* in *Artemia* during the enrichment with *P. tricornutum* from initially axenic (CCMP, (**a**)) or initially non-axenic (IIM, (**b**)) microalgae strains co-cultured with *Ruegeria* sp. ALR6 (RU) or without *Ruegeria* sp. ALR6 (Control). (✕): Total bacteria in RU; (+): Total bacteria in Control; Total *Vibrionaceae* bacteria in RU (▲) and in Control (△). Each point represents the average (*n* = 2) of the concentration (in Log (cells or CFU) mL^−1^) of the microalgae and bacteria. Error bars represents the standard deviation. Graph made with Microsoft Excel 2019.

**Figure 8 microorganisms-11-02715-f008:**
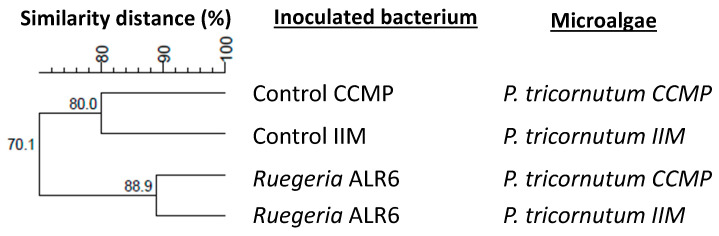
Dendrogram of DGGE profiles of bacterial communities in the *Artemia* enrichment samples. The dendrogram was elaborated based on Dice’s similarity coefficients and the clustering performed by the UPGMA algorithm included in the BioNumerics v7.1 software. The samples analyzed are the ones from the assay of *Artemia* enriched for 48 h with *P. tricornutum* (CCMP and IIM) inoculated with *Ruegeria* sp. ALR6 and the control samples without inoculation. The graph was made by BioNumerics 7.1 software.

## Data Availability

Data is contained within the article or Appendix A.

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
