# Peer review of "Co-Culturing Microalgae with Roseobacter Clade Bacteria as a Strategy for Vibrionaceae Control in Microalgae-Enriched Artemia"

_microorganisms, 2023, doi:10.3390/microorganisms11112715_

Round 1

Reviewer 1 Report

Comments and Suggestions for Authors

This article investigates the co cultivation of different antagonistic bacterial strains with two microalgae commonly used in fish larval culture to enrich prey (such as artemia), enabling a mixed culture of antagonistic bacteria and microalgae to be used as a strategy for bacterial control in aquaculture systems, thereby improving the survival rate of fish larvae. This is a very practical research. There are several questions:

1. How can we confirm that probiotics can colonize microalgae in sufficient quantities? Because the amount of colonization will have an impact on the subsequent antagonistic effect.

2. Probiotics are not dominant bacteria in the aquaculture system, as microalgae containing probiotics enter the artemia aquaculture system. How to ensure the survival of probiotics in aquaculture systems and their effective antagonistic effects against Vibrio?

Author Response

Background has been improved. New contributions have been made to the background in order to improve it and better introduce research (Line 29-41).

Answers to points:

  1. How can we confirm that probiotics can colonize microalgae in sufficient quantities? Because the amount of colonization will have an impact on the subsequent antagonistic effect.

This is a very interesting question, and it has indeed been addressed in the experiment involving the upscaling of the co-culture of microalgae and probiotic bacteria (Figure 6) and subsequent artemia enrichment (Figure 9). During these experiments, we noted that when Ruegeria sp. ARL6 was present at concentrations between 5-6 Log(CFU·mL-1) in the microalgae P. tricornutum co-culture, there was a noticeable decrease in Vibrionaceae counts (as depicted by triangles in Figure 7). Vibrionaceae often include genera that can be potentially harmful to fish, including pathogenic or opportunistic bacteria.

As a result, we could suggest that a concentration of approximately 6 Log(CFU·mL-1) of Ruegeria sp. ARL6 might be adequate to observe the desired probiotic effect. Nevertheless, to substantiate these findings, it would be beneficial to conduct infection assays in the future. These assays would provide a more robust verification of the observed probiotic impact on microalgae and its potential in controlling fish-pathogenic organisms. These last thoughts have been added in the Conclusions (Line 494-497).

  1. Probiotics are not dominant bacteria in the aquaculture system, as microalgae containing probiotics enter the artemia aquaculture system. How to ensure the survival of probiotics in aquaculture systems and their effective antagonistic effects against Vibrio?

We appreciate the observations of the reviewer; however, the aim of this study is to control possible Vibrio in Artemia to avoid fish-larvae mortalities in first feeding with this live prey. One way to ensure the survival of probiotics in rearing systems would be to use green water, with microalgae colonized by probiotics. These thoughts have been added in the Conclusions (Line 499-502)

Reviewer 2 Report

Comments and Suggestions for Authors

Introduction: - The rates and causes of different deaths in larvae should be detailed. - The role of rotifers or artemia in larval nutrition should be clarified.

Results: Are there differences in the chemical composition of algae after bacterial treatments?

Figures: The figure layout is not suitable as it is over text. Check again.

Comments on the Quality of English Language

Minor editing

Author Response

- Background can be improved. 

Answer: New contributions have been made to the background in order to improve it and better introduce research (Line 29-41).

- Are the results clearly presented? To be improved.

Answer: To ensure that we make the necessary enhancements, we kindly request some additional details or specific points that the reviewer 2 believes require improvement in the results section.

Introduction:

- The rates and causes of different deaths in larvae should be detailed 

Answer: This has been clarified in Lines 29-41 of Introduction.

- The role of rotifers or artemia in larval nutrition should be clarified.

Answer: This has been clarified in Lines 29-41 and Lines 63-77 of Introduction.

Results:

- Are there differences in the chemical composition of algae after bacterial treatments?

Answer: The reviewer's question is relevant, particularly because previous studies have noted that bacteria, such as those belonging to the Ruegeria genus, can indeed trigger distinct gene expression patterns in associated diatoms. This phenomenon suggests a potential alteration in the microalgae's metabolite production. This mutualistic interaction carries significant implications, especially in the context of aquaculture. In our research we observed that Ruegeria ALR6 had a significant increase in the growth in the co-culture of Phaeodactylum, (Figure 2h and Table S7) which could be attributed to a mutualistic interaction previously mentioned. Exploring this topic further in future research would yield valuable insights and findings of great interest. These ideas have been included in the Discussion (Line 479-488)

Figures:

- The figure layout is not suitable as it is over text. Check again.

Answer: Spaces has been added between the figures and the captions